# Knowledge-Augmented Language Models for Cause-Effect Relation Classification

**Pedram Hosseini**[1]    **David A. Broniatowski**[1]    **Mona Diab**[1,2]
[1]The George Washington University    [2]Meta AI
phosseini@gwu.edu

## Abstract

Previous studies have shown the efficacy of knowledge augmentation methods in pretrained language models. However, these methods behave differently across domains and downstream tasks. In this work, we investigate the augmentation of pretrained language models with knowledge graph data in the cause-effect relation classification and commonsense causal reasoning tasks. After automatically verbalizing triples in ATOMIC$_{20}^{20}$, a wide coverage commonsense reasoning knowledge graph, we continually pretrain BERT and evaluate the resulting model on cause-effect pair classification and answering commonsense causal reasoning questions. Our results show that a continually pretrained language model augmented with commonsense reasoning knowledge outperforms our baselines on two commonsense causal reasoning benchmarks, COPA and BCOPA-CE, and a Temporal and Causal Reasoning (TCR) dataset, without additional improvement in model architecture or using quality-enhanced data for fine-tuning.

## 1 Introduction

Automatic extraction and classification of causal relations in text has been an important yet challenging task in natural language understanding. Early methods in the 80s and 90s (Joskowicz et al., 1989; Kaplan and Berry-Rogghe, 1991; Garcia et al., 1997; Khoo et al., 1998) mainly relied on defining hand-crafted rules to find cause-effect relations. Starting 2000, machine learning tools were utilized in building causal relation extraction models (Girju, 2003; Chang and Choi, 2004, 2006; Blanco et al., 2008; Do et al., 2011; Hashimoto et al., 2012; Hidey and McKeown, 2016). Word-embeddings and Pretrained Language Models (PLMs) have also been leveraged in training models for understanding causality in language in recent years (Dunietz et al., 2018; Pennington et al., 2014; Dasgupta et al., 2018; Gao et al., 2019).

Investigating the true capability of pretrained language models in understanding causality in text is still an open question. More recently, Knowledge Graphs (KGs) have been used in combination with pretrained language models to address commonsense reasoning. Two examples are Causal-BERT (Li et al., 2020) for guided generation of Cause and Effect and the model introduced by Guan et al. (2020) for commonsense story generation.

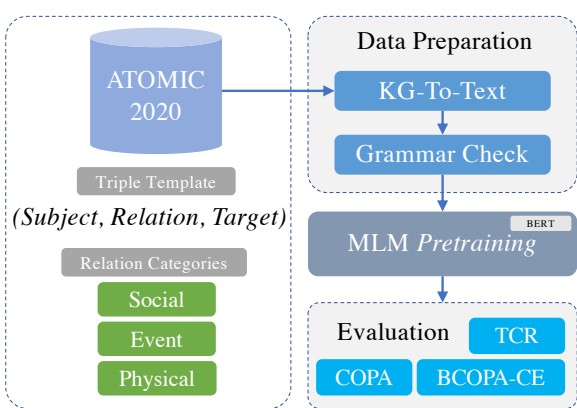

Figure 1: Overview of our proposed framework to continually pretrain PLMs with commonsense reasoning knowledge.

Motivated by the success of continual pretraining of PLMs for downstream tasks (Gururangan et al., 2020), we explore the impact of common sense knowledge injection as a form of continual pretraining for causal reasoning and *cause-effect* relation classification. It is worth highlighting that even though there are studies to show the efficacy of knowledge injection with continual pretraining for commonsense reasoning (Guan et al., 2020), performance of these techniques is very dependent on the domain and downstream tasks (Gururangan et al., 2020). And, to the best of our knowledge, there are limited studies on the effect of commonsense knowledge injection with knowledge graph data on cause-effect relation classification (Dalal

et al., 2021). Our contributions are as follows:

- We study performance of PLMs augmented with knowledge graph data in the less investigated cause-effect relation classification task.

- We demonstrate that a simple masked language modeling framework using automatically verbalized knowledge graph triples, without any further model improvement (e.g., new architecture or loss function) or quality enhanced data for fine-tuning, can significantly boost the performance in cause-effect pair classification.

- We publicly release our knowledge graph verbalization codes and continually pretrained models.

## 2 Method

The overview of our method is shown in Figure 1.[1] We first convert triples in ATOMIC$^{20}_{20}$ (Hwang et al., 2021) knowledge graph to natural language texts. Then we continually pretrain BERT using Masked Language Modeling (MLM) and evaluate performance of the resulting model on different benchmarks. Samples in ATOMIC$^{20}_{20}$ are stored as triples in the form of *(head/subject, relation, tail/target)* in three splits including train, development, and test. ATOMIC$^{20}_{20}$ has 23 relation types that are classified into three categorical types including commonsense relations of social interactions, physical-entity commonsense relations, and event-centric commonsense relations. In the rest of the paper, we refer to these three categories as social, physical, and event, respectively.

### 2.1 Filtering Triples

We remove all duplicates and ignore all triples in which the target value is *none*. Moreover, we ignore all triples that include a blank. Since in masked language modeling we need to know the gold value of masked tokens, a triple that already has a blank (masked token/word) in it may not help our pretraining. For instance, in the triple: `[PersonX affords another ___,` `xAttr, useful]` it is hard to know why or understand what it means for a person to be useful without knowing what they afforded. This preprocessing step yields in 782,848 triples with 121,681,

177,706, and 483,461 from event, physical, and social categories, respectively. Distribution of these relations is shown in Figure 2.

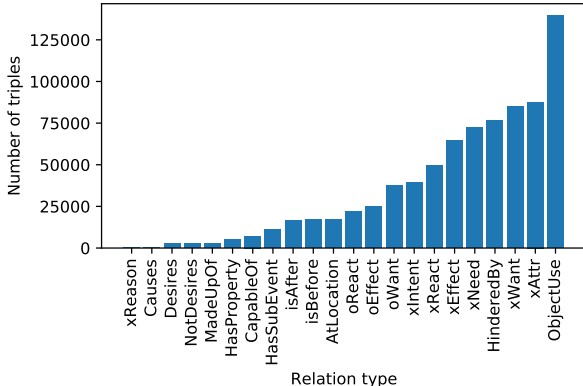

Figure 2: Distribution of relation types in ATOMIC$^{20}_{20}$.

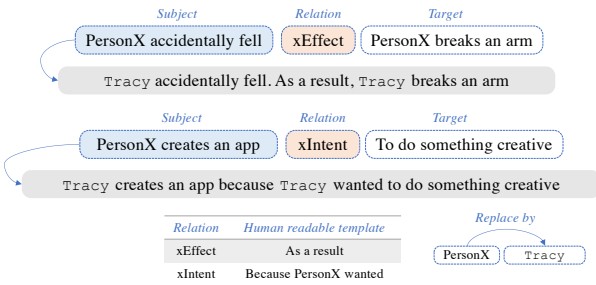

Figure 3: Examples of converting two triples in ATOMIC$^{20}_{20}$ to natural language text using human readable templates. Following Sap et al. (2019), we replace *PersonX* with a name.

### 2.2 Converting Triples

Each relation in ATOMIC$^{20}_{20}$ is associated with a human-readable template. For example, *xEffect*'s and *HasPrerequisite*'s templates are *as a result, PersonX will* and *to do this, one requires*, respectively. We use these templates to convert triples in ATOMIC$^{20}_{20}$ to sentences in natural language by concatenating the subject, relation template, and target. Examples of converting triples to text are shown in Figure 3.

### 2.3 Checking Grammar

When we convert triples to natural language text, ideally we want to have grammatically correct sentences. Human readable templates provided by ATOMIC$^{20}_{20}$ are not necessarily rendered in a way to form error-free sentences when concatenated with subject and target in a triple. To address this issue, we use an open-source grammar and spell

---

[1]Codes and models are publicly available at https://github.com/phosseini/causal-reasoning.

checker, LanguageTool,[2] to double-check our converted triples to ensure they do not contain obvious grammatical mistakes or spelling errors. Similar approaches that include deterministic grammatical transformations were also previously used to convert KG triples to coherent sentences (Davison et al., 2019). It is worth pointing out that the Data-To-Text generation (KG verbalization) for itself is a separate task and there have been efforts to address this task (Agarwal et al., 2021). We leave investigating the effects of using other Data-To-Text and grammar-checking methods to future research.

## 2.4 Continual Pretraining

As mentioned earlier, we use MLM to continually pretrain our PLM, *BERT-large-cased* (Devlin et al., 2018). We follow the same procedure as BERT to create the input data to our pretraining (e.g., number of tokens to mask in input examples). We run the pretraining using ATOMIC$_{20}^{20}$'s *train* and *development* splits as our training and evaluation sets, respectively, for 10 epochs on Google Colab TPU v2 using *PyTorch/XLA* package with a maximum sequence length of 30 and batch size of 128.[3] To avoid overfitting, we use early stopping with the patience of 3 on evaluation loss. We select the best model based on the lowest evaluation loss at the end of training.[4]

## 3 Experiments

### 3.1 Benchmarks

We chose multiple benchmarks of commonsense causal reasoning and cause-effect relation classification to ensure we thoroughly test the effects of our newly trained models. These benchmarks include: 1) Temporal and Causal Reasoning (TCR) dataset (Ning et al., 2018), a benchmark for joint reasoning of temporal and causal relations; 2) Choice Of Plausible Alternatives (COPA) (Roemmele et al., 2011) dataset which is a widely used and notable benchmark (Rogers et al., 2021) for commonsense causal reasoning; And 3) BCOPA-CE (Han and Wang, 2021), a new benchmark inspired by COPA, that contains unbiased token distributions which makes it a more challenging benchmark. For COPA-related experiments, since COPA does not have a training set, we use COPA's

development set for fine-tuning our models and testing them on COPA's test set (COPA-test) and BCOPA-CE. For hyperparameter tuning, we randomly split COPA's development set into train (%90) and dev (%10) and find the best learning rate, batch size, and number of train epochs based on the evaluation accuracy on the development set. Then using COPA's original development set and best set of hyperparameters, we fine-tune our models and evaluate them on the test set. In all experiments, we report the average performance of models using four different random seeds. For TCR, we fine-tune and evaluate our models on train and test splits, respectively.

## 3.2 Models and Baseline

We use *bert-large-cased* pre-trained model in all experiments as our baseline. For COPA and BCOPA-CE, we convert all instances to a SWAG-formatted data (Zellers et al., 2018) and use Huggingface's *BertForMultipleChoice* –a BERT model with a multiple-choice classification head on top. And for TCR, we convert every instance by adding special tokens to input sequences as event boundaries and use the R-BERT [5] model (Wu and He, 2019). We chose R-BERT for our relation classification since it not only leverages the pretrained embeddings but also transfers information of target entities (e.g., events in a relation) through model's architecture and incorporates encodings of the targets entities. Examples of COPA and TCR are shown in Figure 4. BCOPA-CE has the same format as COPA.

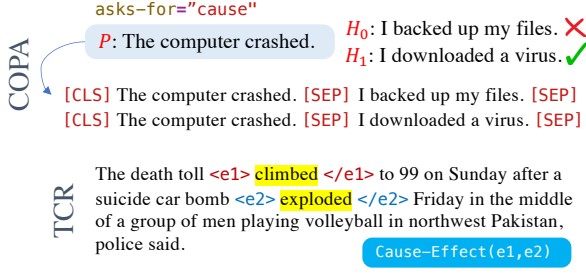

Figure 4: COPA and TCR examples. The COPA instance is converted to Multiple Choice format.

## 4 Results and Discussion

Results of our experiments on TCR are shown in Table 1. As can be seen, our model significantly outperforms both our baseline and the joint infer-

---

[2]https://tinyurl.com/yc77k3fb

[3]%99.99 of ATOMIC$_{20}^{20}$ instances have 30 tokens or less.

[4]We use Huggingface's *BertForMaskedLM* implementation.

[5]We use the following implementation of R-BERT: https://github.com/monologg/R-BERT

ence framework by Ning et al. (2018) formulated as an integer linear programming (ILP) problem.

| Model | Acc (%) |
|---|---|
| Joint system (Ning et al., 2018) | 77.3 |
| BERT-large (baseline) ✳ | 75.0 |
| ATOMIC-BERT-large$_{MLM}$ ✳ | **91.0** |

Table 1: TCR Accuracy results. ✳ Our models

Results of experiments on COPA-test are shown in Table 2. We initially observed that a continually pretrained model using all three types of relations has a lower performance than our baseline. By taking a closer look at each relation type, we decided to train another model, this time only using the *event* relations. The reason is that event-centric relations in ATOMIC$_{20}^{20}$ specifically contain commonsense knowledge about event interaction for understating likely causal relations between events in the world (Hwang et al., 2021). In addition, event relations have a relatively longer context (# of tokens) than the average of all three relation types combined which means more context for a model to learn from. Our new pretrained model outperformed the baseline by nearly %5 which shows the effect of augmented pretrained language model with commonsense reasoning knowledge.

| Model | Acc (%) |
|---|---|
| PMI (Roemmele et al., 2011) | 58.8 |
| b-l-*reg* (Han and Wang, 2021) | 71.1 |
| Google T5-base (Raffel et al., 2019) | 71.2 |
| BERT-large (Kavumba et al., 2019) | 76.5 |
| CausalBERT (Li et al., 2020) | 78.6 |
| BERT-SocialIQA (Sap et al., 2019)* | 80.1 |
| BERT-large (baseline) ✳ | 74.4 |
| ATOMIC-BERT-large$_{MLM}$ ✳ | |
|   - Event only | 79.2 |
| Google T5-11B (Raffel et al., 2019) | 94.8 |
| DeBERTa-1.5B (He et al., 2020) | 96.8 |

Table 2: COPA-test Accuracy results. ✳ Our models. * For a fair comparison, we report BERT-SocialIQA's average performance.

We further experiment on the *Easy* and *Hard* question splits in COPA-test separated by Kavumba et al. (2019) to see how our best model performs on harder questions that do not contain superficial cues. Results are shown in Table 3. As can be seen, our ATOMIC-BERT model significantly outperforms both the baseline and former models on Hard and Easy questions.

| Model | Easy ↑ | Hard ↑ |
|---|---|---|
| (Han and Wang, 2021) | - | 69.7 |
| (Kavumba et al., 2019) | 83.9 | 71.9 |
| BERT-large (baseline) ✳ | 83.0 | 69.2 |
| ATOMIC-BERT-large ✳ | **88.9** | **73.1** |

Table 3: COPA-test Accuracy results on Easy and Hard question subsets. ✳ Our models.

It is worth mentioning three points here. First, our model, BERT-large, has a significantly lower number of parameters than state-of-the-art models, Google T5-11B (∼32x) and DeBERTa-1.5B (∼4x) and it shows how smaller models can be competitive and benefit from continual pretraining. Second, we have not yet applied any model improvement methods such as using a margin-based loss introduced by Li et al. (2019) and used in Causal-BERT (Li et al., 2020), an extra regularization loss proposed by Han and Wang (2021), or fine-tuning with quality-enhanced training data, BCOPA, introduced by Kavumba et al. (2019). As a result, there is still great room to improve current models that can be a proper next step. Third, we achieved a performance on par with BERT-SocialIQA (Sap et al., 2019) [6] while we did not use crowdsourcing or any *manual* re-writing/correction, which is expensive, for verbalizing KG triples to create our pretraining data.

| Model | Acc (%) |
|---|---|
| b-l-*aug* (Han and Wang, 2021) | 51.1 |
| b-l-*reg* (Han and Wang, 2021) | 64.1 |
| BERT-large (baseline) ✳ | 55.8 |
| ATOMIC-BERT-large$_{MLM}$ ✳ | |
|   - Event only | 58.1 |

Table 4: BCOPA-CE Accuracy results. ✳ Our models. * Base model in *b-l-*\* is BERT-large.

### 4.1 BCOPA-CE: Prompt vs. No Prompt

Results of experiments on BCOPA-CE are shown in Table 4. As expected based on the results also reported by Han and Wang (2021), we initially observed that our models are performing nearly as random baseline. Since we do not use the type of question when encoding input sequences, we decided to see whether adding question type as a prompt to input sequences will improve the performance. We added `It is because` and `As a`

---

[6]Our best random seed run achieved %81.4 accuracy.

result, as prompt for `asks-for="cause"` and `asks-for="effect"`, respectively. Interestingly, the new model outperforms the baseline and [Han and Wang (2021)](#)'s *b-l-aug* model that is fine-tuned with the same data as ours, when question types are added as prompts to input sequences of correct and incorrect answers in the test set. We also ran a similar experiment on COPA-test (Table 5) in which adding prompt did not help with performance improvement.

| Train / Test | ✗ Prompt | ✓ Prompt |
|---|---|---|
| ✗ Prompt | **79.2** | 76.4 |
| ✓ Prompt | 75.5 | 77.9 |

Table 5: COPA-test Accuracy ablation study results for prompt vs. no prompt.

## 5   Conclusion

We introduced a simple framework for augmenting PLMs with commonsense knowledge created by automatically verbalizing ATOMIC$_{20}^{20}$. Our results show that commonsense knowledge-augmented PLMs outperform the original PLMs on cause-effect pair classification and answering commonsense causal reasoning questions. As the next step, it would be interesting to see how the previously proposed model improvement methods or using unbiased fine-tuning datasets can potentially enhance the performance of our knowledge-augmented models.

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
