# OpenReview forum: "Knowledge-Augmented Language Models for Cause-Effect Relation Classification"
_aclweb.org/ACL/2022/Workshop/CSRR — ACL 2022 Workshop CSRR_

### Official Review · Reviewer_327F · 2022-03-19

**Rating:** 6
**Confidence:** 4

**Review:**

Summary
- The technique of continually pretraining language models on commonsense knowledge graph triples has been shown useful for some downstream tasks, but it may depend on the specific domain and tasks. This work investigates the effect of this technique on the task of cause-effect relation classification. The authors verbalize the ATOMIC2020 knowledge graph and continue to pretrain BERT-large on it. The authors show that this simple method can boost the performance in cause-effect classification.

Reasons to Accept:
- The effect of commonsense knowledge graphs for cause-effect relation classification is an interesting topic, but has not been studied systematically. This work performs an interesting investigation into this research question.
- The paper is clear and well-written overall.
- The authors will publicly release the knowledge graph verbalization codes and the trained models

Weakness and questions:
- Overall, I think that the experiments/analyses could be polished a bit more. Below are s few suggestions.
- ATOMIC-BERT-large (Event, Physical, Social relations) underperforms the baseline BERT-large on two datasets (Table 2, 4). It'd be great if the authors could investigate more into why this is the case. Do Physical/Social relations have very different distributions of knowledge than the tasks of interest (i.e. cause-effect prediction)? Even though ATOMIC-BERT-large (Event) outperforms the baseline, it is a bit concerning that in order for the proposed method to work, it needs to identify what kinds of relations within ATOMIC2020 is useful or hurting for the task and remove the hurting relations. It'd be ideal if the authors could think about a bit more elegant method to address this issue.
- Additionally, it is not clear why adding prompt helps for BCOPA-CE but hurts for COPA task. It'd be great if the authors could conduct more in-depth analysis for their results.

Typos/grammar:
- L184: redundant parenthesis in "(MLM)"?

---

### Official Review · Reviewer_G1op · 2022-03-23
**Unique limited contribution of cause-effect models created by knowledge-augmented LM pretraining**

**Rating:** 5
**Confidence:** 5

**Review:**

This paper proposes a method for knowledge-augmented LM pretraining with cause-effect information. The method is targetted towards causal reasoning benchmarks (TCR, COPA, and BCOPA-CE). The method performs better than vanilla and existing system baselines on TCR, and below some baselines on the COPA/BCOPA-CE tasks.

The paper is overall interesting. Its novelty is limited as the method is already known in the literature, but the evaluation is unique which may be enough for a workshop paper.

Weaknesses:
* It is unclear how the citations in paragraph 1 of section 1 relate to the statement that PLMs have been leveraged in for understanding causality in language.
* The statement that model performance is very dependent on the domain and downstream tasks is reasonable, but it is broad, and it is unclear how this paper addresses this challenge.
* The evaluation contains various benchmark-specific adaptations which are not anticipated in the experimental setup, and feel like hacks to improve performance ad hoc. It would be good to give these configurations a better structure in the paper, ideally by stating them within the method description or within the experimental setup. Moreover, it would be good to clarify how each of these configurations relates to the research question investigated in this paper.

---

### Official Review · Reviewer_Yi8h · 2022-03-23
**Review of the paper**

**Rating:** 4
**Confidence:** 3

**Review:**

# Summary
This work investigates the augmentation of pretrained language models (LMs) with knowledge graphs (KGs) for the cause-effect relation classification and commonsense causal reasoning tasks. They verbalize the ATOMC-2020 KG triples into natural language which they use to continually pretrain BERT. Their results show that the continually pretrained LM outperforms non-continually pretrained ones on two commonsense causal reasoning benchmarks, COPA and BCOPA-CE, and a Temporal and Causal Reasoning (TCR) dataset.

# Contributions
1. They study pretrained LMs augmented with the ATOMIC-2020 knowledge graph in the commonsense reasoning domain.
2. They perform experiments to show that these augmented LMs can outperform non-continually pretrained ones and other baselines on the cause-effect relation classification and commonsense causal reasoning tasks.

# Pros
1. The writing is generally very clear, which makes the paper easy to follow.
2. The result on the TCR task looks very good!
3. Approach the (causal) commonsense reasoning task, which is very important.

# Cons
1. The framework of continually pertaining LMs using verbalized KG triples is something that has been done previously [1]. The only things that are different in this paper is to apply this technique to a different KG (ATOMIC-2020) and to fine-tune on a few different tasks and benchmarks. So there is a lack of novelty.
2. I find the result in Table 4 unsatisfactory. First, what is the b-l-reg baseline and why does the ATOMIC-BERT model underperform that baseline? Second, the fact that using all the categories for ATOMIC-2020 actually hurt performance but only using the event ones does not fit well with the claim of the paper that general commonsense knowledge helps the causal commonsense reasoning task. It may just be the case that the event triples in ATOMIC-2020 is in a closer domain to the BCOPA-CE and it is actually the in-domain further pertaining that is helping. Third, why not just try using the causal relations in ATOMIC-2020 ("cause", "effect" etc)?
3. The standard deviations are not reported for all the experimental results.
4. I know it is not a fair comparison to compare ATOMIC-BERT and T5 and DeBERTa, but looking at the latter two's numbers on the COPA-test, the task seems a solved one. I am not sure how significant/useful it is to continue working on this benchmark.

# Other comments and questions
1. An ablation study in the effect of different ways to verbalize the KG triples and e.g. whether the grammar correction step is necessary can be useful and interesting.
2. Which split of ATOMIC-2020 is used?

# References
1. Guan, Jian, et al. "A knowledge-enhanced pretraining model for commonsense story generation." Transactions of the Association for Computational Linguistics 8 (2020): 93-108.

---

### Decision · Program_Chairs · 2022-03-28

Accept